# Educational Stress among Greek Adolescents: Associations between Individual, Study and School-Related Factors

**DOI:** 10.3390/ijerph20064692

**Published:** 2023-03-07

**Authors:** Evangelia Moustaka, Flora Bacopoulou, Kyriaki Manousou, Christina Kanaka-Gantenbein, George P. Chrousos, Christina Darviri

**Affiliations:** 1Postgraduate Course of Science of Stress and Health Promotion, School of Medicine, National and Kapodistrian University of Athens, 11527 Athens, Greece; 2University Research Institute of Maternal and Child Health & Precision Medicine and UNESCO Chair on Adolescent Health Care, National and Kapodistrian University of Athens, Aghia Sophia Children’s Hospital, 11527 Athens, Greece; 3First Department of Pediatrics, Medical School, National and Kapodistrian University of Athens, Aghia Sophia Children’s Thivon & Papadiamantopoulou St., 11527 Athens, Greece

**Keywords:** educational stress, adolescents, students, Greece, school, academic

## Abstract

Greek adolescent students experience high levels of educational stress during the school period. In this cross-sectional study, we examined various factors associated with educational stress in Greece. The study was conducted using a self-report questionnaire survey in Athens, Greece, between November 2021 and April 2022. We studied a sample of 399 students (61.9% females; 38.1% males, with a mean age of 16.3 years). We found that several factors, such as age, sex, study hours, and health status of the adolescents, were associated with the subscales of the Educational Stress Scale for Adolescents (ESSA), Adolescent Stress Questionnaire (ASQ), Rosenberg Self-Esteem Scale (RSES), and State-Trait Anxiety Inventory (STAI). Older age, female sex, family status, parental profession, and the number of study hours were positively associated with the amount of stress, anxiety and dysphoria-related symptoms reported by the students, including pressure from studying, worry about grades, and despondency. Future research studies are needed to aid specialized interventions for adolescent students to effectively deal with their academic challenges.

## 1. Introduction

Students’ educational stress and anxiety are multidimensional and are linked to various aspects of the school environment and, generally, education itself [1]. Academic performance plays a key part in youths’/adolescents’ decision-making processes regarding their next, higher level of education and, probably, their career and adult life as well [2]. Indeed, in the educational process, a significant number of students experience stress and underperform during the school years or during the examination periods; they may also have a reduced sense of self-efficacy and even engage in negative self-evaluation [3].

Research in this area has suggested that the most disturbing daily hassles causing educational stress usually include: constant pressure to study, time limitations, homework assigned by teachers, and the actual process of the examinations. Adolescent students also experience high stress levels because of their own academic expectations but also because of their teachers’ and family members’ expectations [4]. High stress levels during the examination periods and the education process itself are linked to academic underperformance and low self-esteem of some adolescents [5]. Conversely, low stress and anxiety levels are linked to better academic performance during the examination periods.

The scientific literature and related research on sex differences in adolescence show statistically significant differences between males and females in the degree and intensity of dysphoria they experience in response to various stressors [6]. In general, adolescent girls experience more emotional difficulties than boys, including anxiety and depression [6], and these differences are also observed in educational stress, with adolescent girls reporting higher levels of stress dysphoria than boys [7].

Potential stressors in the school environment include problematic interaction with peers or teachers, school performance demands, school rules, and conflict between time allocated for reading and other extracurricular activities [8]. In addition, studies show a significant increase in the school pressure experienced by adolescents in different countries [9]. This increasing trend is alarming because school pressure often results in or is intertwined with the “typical” stress frequently experienced by adolescent students, that negatively impacts their mental and physical health and educational outcomes [10].

For decades, the Greek society has given special consideration to education and value to higher education. Education and admission to a school of higher education is a key step in the life of a Greek teenager, and important for the social, economic and personal development of the individual [11]. This perception has clearly influenced and determined the direction of the Greek educational system at various levels. Undoubtedly, the panHellenic exams, the entrance exams to higher education, are a milestone in the Greek educational system [12]. For the majority of teenagers, the preparation period for the panHellenic exams, which begins long before the 3rd grade of Lyceum, is a dynamic and gradual source of stress and anxiety since admission to higher education is a prerequisite for further professional development and reflects the transition to adulthood. Adolescent students are faced with physical and psychological challenges during their preparation, as they are asked to make critical and important decisions for their future career path, while they need to manage potential pressure and conflict with parents due to the high demands of the latter for academic and social success and recognition [11]. In particular, the last two grades of high school, where the preparation for admission to higher education culminates, are a period of intense anxiety, stress and worry for young students, testing their endurance and strength [13].

In this study, we examined various associations between educational stress, sex and lifestyle factors, as well as the roles of parents and study and leisure characteristics in the lives of adolescent students.

## 2. Materials and Methods

### 2.1. Research Sample Description 

A cross-sectional study was conducted with a self-report questionnaire survey in Greece, in Athens, between November 2021 and April 2022.

A sample of 399 students (61.9% female; 38.1% male), with a mean age of 16.3 years, participated in this study. The process of completing the questionnaires took about 20 min. Participation in the survey was voluntary, no payment of any kind was given to the participants, and the anonymity and confidentiality of the responses of the participants were ensured. The sample was community-sourced and administered online through the Google Forms platform. Eligible students were adolescents aged 12 to 18 years, able to write and read in the Greek language, who provided a written, signed consent form by a parent or legal guardian.

The research protocol was approved by the Committee of Research and Ethics of the National and Kapodistrian University of Athens, School of Medicine (Ref. 93776, 11 August 2021).

### 2.2. Measurements and Data Collection Tools

For the purposes of the research, a demographic questionnaire was initially given with reference to the sex of the participants, age and school class, as well as the educational and professional level of the parents. It also included questions about students’ studies, hours spent on the Internet or free time, and questions about adolescents’ lifestyles and habits, adapted from Sun et al. [2].

Educational Stress Scale for Adolescents (ESSA): This questionnaire includes 16 items/questions aimed at assessing educational stress. Each question consists of a 5-point Likert scale (ranging from 1 = strongly disagree to 5 = strongly agree). The participants’ responses to the questions add up to a total score; moreover, 5 different factors resulted from their statistical analysis: ‘Pressure from study’; ‘Workload’; ‘Worry about grades’; ‘Self-expectation’, and ‘Despondency’. Higher scores indicate greater stress. Published in English by Sun et al. (2011), the questionnaire has good internal consistency, with Cronbach’s *α* = 0.82 for the total scale, and *α* = 0.79, *α* = 0.73, *α* = 0.69, *α* = 0.65, and *α* = 0.64 for the five factors, respectively [2].

Adolescent Stress Questionnaire—ASQ: This questionnaire, adapted in Greek by Darviri et al. (2014), measures various situations and events in the life of adolescents that cause stress with a 5-point Likert scale (1 = not at all stressful to 5 = very stressful). The internal validity of this questionnaire, as assessed with Cronbach’s alpha, is in the range of 0.70–0.86 and has a general value of 0.96 [12].

Rosenberg Self-Esteem Scale: This questionnaire measures self-esteem with a 4-point Likert scale (1 = strongly agree to 4 = strongly disagree) with a high reliability (α = 0.809). A higher score on the scale indicates higher levels of self-esteem [13].

*State-Trait Anxiety Inventory (STAI):* This questionnaire comprises of two subscales measuring state and trait anxiety. Cronbach’s *α* was 0.93 for the state anxiety subscale and 0.92 for the trait anxiety subscale. Higher levels on the scale indicate greater stress [14].

## 3. Results

### 3.1. Demographic Characteristics

The research sample’s demographic characteristics are presented in Table 1. A total of 399 students (61.9% females) participated in the study, with a mean (SD) age of 16.3 (1.4) years. Most participants had siblings (81.2%). The majority of the participants’ parents were married (72.9%) and held a university/technical educational institute degree or a Master’s degree/Ph.D. (68.2%).

Most students reported spending more than 2 h per day on schoolwork (320, 80.2%) and having private tutoring (349, 87.5%) classes, as well as during the weekend or holidays (204, 51.1%), and high or very high grades (261, 65.4%) during the last year.

Regarding the period of the last 30 days prior to their inclusion in the study, most participants reported having a good health status (294, 73.7%), participating in physical exercise (309, 77.4%), using the Internet (398, 99.7%), and playing electronic games (214, 54.6%).

### 3.2. Study Measurements

The median (IQR) and mean (SD) scores for all study measurements are presented in Table 2. The mean (SD) score was 47.7 (10.0) for the ESSA total score and 12.0 (4.0), 9.9 (2.5), 8.4 (2.5), 9.5 (2.7) and 8.0 (2.3) for the pressure from study, workload, worry about grades, self-expectation and despondency subscales, respectively. The mean (SD) STAI-state, STAI-trait and self-esteem scale scores were 44.0 (15.7), 44.5 (12.7) and 27.4 (5.3), respectively. Finally, the mean (SD) scores for the ASQ scales were 35.8 (10.8) for the stress of home life subscale, 22.4 (6.2) for the stress of school performance, 8.3 (3.3) for the stress of school attendance, 15.4 (6.1) for the stress of romantic relationships, 20.5 (7.7) for the stress of peer pressure, 18.8 (7.1) for the stress of teacher interaction, 10.9 (3.1) for the stress of future uncertainty, 17.1 (4.7) for the stress of school/leisure conflict, 11.9 (4.3) for the stress of financial pressure, and 8.1 (3.1) for the stress of emerging adult responsibility.

### 3.3. Associations between Demographic Characteristics and Examined Scales

The differences in ESSA scales by selected demographic characteristics, as well as the correlations (Spearman’s rho) with the rest of the scales examined, are presented in Table 3.

### 3.4. Demographic Characteristics

Female students had higher scores than male students in all scales, and all the differences were statistically significant (‘Self-expectation’: *p* < 0.001; ‘Workload’: *p* = 0.002; ‘Despondency’: *p* = 0.014; ‘Pressure from study’: *p* < 0.001; ‘Worry about grades’: *p* < 0.001). There was a statistically significant, positive correlation between age and all the ESSA scales; nevertheless, all the correlations were of low magnitude, the strongest among them being the one with the ‘Pressure from study’ scale (rho = 0.220; *p* < 0.001). Non-statistically significant differences were observed in the ESSA scales between students with and without siblings, the sole exception being the ‘Despondency’ scale, where students with siblings had a slightly lower mean compared to those without (*p* = 0.017).

There was a statistically significant difference in terms of parental family status in the ‘Worry about grades’ scale, with the students who had lost at least one parent having the highest mean score (*p* = 0.004); moreover, in terms of parental work status, there was a significant difference in the ‘Despondency’ scale, where students with unskilled parents had the highest mean score (*p* = 0.007).

### 3.5. Study- and Leisure-Related Characteristics

The number of hours the students spent studying per day over the past 12 months seems to be associated with all the ESSA scales, with the students that spent over 3 h per day on studying having the highest scores on all scales (‘Self-expectation’: *p* < 0.0001; ‘Workload’: *p* < 0.0001; ‘Despondency’: *p* = 0.0001; ‘Pressure from study’: *p* < 0.0001; ‘Worry about grades’: p = 0.002). Likewise, the students who attended private individual or group tutoring classes (‘Self-expectation’: *p* < 0.0001; ‘Workload’: *p* < 0.0001; ‘Despondency’: *p* = 0.001; ‘Pressure from study’: *p* = 0.001; ‘Worry about grades’: *p* = 0.005) or classes during weekends or holidays (*p* < 0.0001 for all scales) over the past 12 months, had statistically higher scores in all the ESSA scales.

The students’ academic grades over the past 12 months were found to be correlated, to a statistically significant extent, with the ‘Despondency’ (*p* < 0.0001) and ‘Worry about grades’ (*p* < 0.0001) scales, with the students that reported lower grades having higher scores in the respective scales (Table 3).

Health status over the past 30 days was associated to a statistically significant extent with all the scales (Table 3: ‘Self-expectation’: *p* < 0.0001; ‘Workload’: *p* < 0.0001; ‘Despondency’: *p* = 0.002; ‘Pressure from study’: *p* < 0.0001; ‘Worry about grades’: *p* = 0.020). Students reporting poor health status had higher scores in the ‘Self-expectation’, ‘Workload’, ‘Despondency’ and ‘Pressure from study’ scales. Furthermore, students reporting engagement in some form of physical exercise over the past 30 days had lower scores in the ‘Workload’ (*p* = 0.001), ‘Despondency’ (*p* = 0.019), and ‘Pressure from study’ (*p* = 0.046) scales.

Playing electronic games over the past 30 days seems to be associated with lower scores in the ‘Self-expectation’ (*p* < 0.0001) and ‘Workload’ (*p* < 0.0001) scales.

### 3.6. Correlations between the Scales Examined

There were statistically significant, positive correlations between the ESSA subscales and those of the STAI and ASQ questionnaires (Table 3). The strongest correlations (rho > 0.6) were observed between the scales of ‘Self-expectation’ and ‘STAI-state’ (rho = 0.627; *p* < 0.001), ‘Self-expectation’ and ‘STAI-trait’ (rho = 0.668; *p* < 0.001), ‘Pressure from study’ and ‘ASQ-Stress of Home Life’ (rho = 0.617; *p* < 0.001), and ‘Pressure from study’ and ‘ASQ-Stress of School Performance’(rho = 0.694; *p* < 0.001). Statistically significant negative correlations were observed between the ESSA subscales and the ‘Self-esteem’ scale (rho range: from −0.270 for the ‘Workload’ scale to −0.574 for the ‘Despondency’ scale).

## 4. Discussion

In this cross-sectional study, we examined a wide range of factors and influences that were related to perceived educational stress among Greek high school students. Many variables had statistically significant relations with some subscales of ESSA. The variables of the latter scale were taken into consideration for the first time regarding educational stress in Greece. The ESSA questionnaire was validated and cross-culturally adapted simultaneously with this study.

Regarding the correlations of demographic characteristics with the scales of the educational stress assessment questionnaire under examination, a very important finding, in line with the existing literature, is that female students had higher scores than male students in all the ESSA scales, and all the differences were statistically significant [9]. A statistically significant, positive correlation of age with all the ESSA scales was also observed. One reason may be that females are more likely to regard school performance as a very important task. Adolescent girls were more concerned about most things than boys. Additionally, they indicated experiencing more stressful events [11]. All these correlations were rather small, the strongest one being observed with the ‘Pressure from study’ scale (rho = 0.220; *p* < 0.001). Our finding that in the ESSA scale, older students experience higher academic stress levels, agrees with the study by Sun et al. [2].

There was a statistically significant difference in terms of parental family status in the ‘Worry about grades’ scale, with the students that had lost at least one parent having the highest mean score (*p* = 0.004); in terms of parental work status, a significant difference was also observed in the ‘Despondency’ scale, in which students with unskilled parents had the highest mean score (*p* = 0.007). This is contrary to the research conducted by Sun et al. (2013), who reported that parental family and work status did not affect the children’s educational stress [11].

Our research showed that the reported number of study hours per day over the past 12 months was related to all the ESSA scales, with the students that spent over 3 h per day on studying having the highest scores in all scales (‘Self-expectation’: *p* < 0.0001; ‘Workload’: *p* < 0.0001; ‘Despondency’: *p* = 0.003; ‘Pressure from study’: *p* < 0.0001; ‘Worry about grades’: p = 0.002). Likewise, students who attended private individual or group tutoring classes (‘Self-expectation’: *p* < 0.0001; ‘Workload’: *p* < 0.0001; ‘Despondency’: *p* = 0.001; ‘Pressure from study’: *p* = 0.001; ‘Worry about grades’: *p* = 0.005) or classes during the weekends or holidays (*p* < 0.0001 for all scales) over the past 12 months had statistically significant higher scores in all the ESSA scales. All of these findings are in line with the existing literature, which has established a strong correlation between students’ study hours and academic stress [11]. Moreover, and, more specifically, students who attended private individual or group tutoring classes also experienced stress and pressure.

An additional important finding is that the students’ academic grades over the past 12 months were correlated with the ‘Despondency’ (*p* < 0.0001) and ‘Worry about grades’ (*p* < 0.0001) scales, with the students that reported lower grades having higher scores in the respective scales. This finding is partly in line with the existing literature, according to which students with low score averages experience higher levels of perceived stress; however, there is currently no research that has shown a correlation between low score averages and specific variables of the educational stress assessment questionnaire.

As suggested in the relevant international literature, adolescent educational stress is linked to physical and mental health problems [14]. Our study showed that the health status of students over the 30 days prior to their participation in the research was correlated, to a statistically significant extent, with all the scales of the educational stress assessment questionnaire (‘Self-expectation’: *p* < 0.0001; ‘Workload’: *p* < 0.0001; ‘Despondency’: *p* = 0.002; ‘Pressure from study’: *p* < 0.0001; ‘Worry about grades’: *p* = 0.020). The students that reported poor health status had higher scores in the ‘Self-expectation’, ‘Workload’, ‘Despondency’, and ‘Pressure from study’ scales. Moreover, the students that reported engagement in some form of physical exercise over the past 30 days had lower scores in the ‘Workload’ (*p* = 0.001), ‘Despondency’ (*p* = 0.019) and ‘Pressure from study’ (*p* = 0.046) scales. This important finding confirms several interventional studies in the adolescent population, showing that exercise plays a key part in reducing the stress that is related to school and to the various manifestations of this developmental stage [10].

This study has some limitations. First of all, the information was collected relying on self-report measurements, and some recall bias cannot be avoided. Second, the relationships between factors cannot be interpreted causally because of the cross-sectional nature of this study. Third, the sample was chosen conveniently, and the findings cannot be generalised to the population, although it should be noted that the demographic characteristics of our sample were similar to the population of similar ages in Greece in terms of sex and place of residence.

## 5. Conclusions

In conclusion, this research showed that many adolescents experience excessive educational stress and poor health status. Future research studies should include a larger sample of adolescents from around the country and examine specific factors that exacerbate educational stress to aid the development of specialized interventional programs for adolescent students to effectively deal with their academic challenges.

## Figures and Tables

**Table 1 ijerph-20-04692-t001:** Sociodemographic characteristics of the study sample (*N* = 399).

**Sex, *N* (%)**	
Males	152 (38.1)
Females	247 (61.9)
**Age (years)**	
Median (IQR)	17.0 (1.0)
Mean (SD)	16.3 (1.4)
**Class, *N* (%)**	
Seventh grade (Gymnasium)	12 (3.0)
Eighth grade (Gymnasium)	30 (7.5)
Ninth grade (Gymnasium)	38 (9.5)
Tenth grade (Lyceum)	94 (23.6)
Eleventh grade (Lyceum)	115 (28.8)
Twelfth grade (Lyceum)	110 (27.6)
**Place of residence, *N* (%)**	
Athens	355 (89.0)
Other	44 (11.0)
**Siblings, *N* (%)**	
Yes	324 (81.2)
No	75 (18.8)
**Parents’ Marital Status, *N* (%)**	
Married	291 (72.9)
Divorced/Separated	100 (25.1)
Death of one or both parents	8 (2.0)
**Parents’ Educational status, *N* (%)**	
Until Upper Secondary School (Lyceum)	127 (31.8)
Bachelor	207 (51.9)
MSc/Ph.D.	65 (16.3)
**Parents’ work status, *N* (%)**	
Unskilled	22 (5.5)
Semi-skilled	28 (7.0)
Skilled	349 (87.5)

**Time spent on schoolwork, daily, *N* (%)**	
Less than 2 h	79 (19.8)
2–3 h	182 (45.6)
More than 3 h	138 (34.6)
**Private tutoring during the last year, *N* (%)**	
Yes	349 (87.5)
No	50 (12.5)
**Class during weekends or holidays, *N* (%)**	
Yes	204 (51.1)
No	195 (48.9)
**Grades during last year, *N* (%)**	
Low	13 (3.3)
Medium	125 (31.3)
High	194 (48.6)
Very high	67 (16.8)
**Health state during the last 30 days, *N* (%)**	
Bad	9 (2.3)
Average	96 (24.1)
Good	294 (73.7)
**Physical exercise during the last 30 days, *N* (%)**	
Yes	309 (77.4)
No	90 (22.6)
**Use of Internet during the last 30 days, *N* (%)**	
Yes	398 (99.7)
No	1 (0.3)
**Electronic games during the last 30 days, *N* (%)**	
Yes	214 (53.6)
No	185 (46.4)

**Table 2 ijerph-20-04692-t002:** Sample’s study measurements (*N* = 399).

Scale and Subscale Scores	Median (IQR) Mean (SD)
ESSA—Total Score	49.0 (15.0) 47.7 (10.0)
ESSA—Pressure from Study	12.0 (4.0) 12.0 (3.0)
ESSA—Workload	10.0 (4.0) 9.9 (2.5)
ESSA—Worry about Grades	8.0 (4.0) 8.4 (2.5)
ESSA—Self-expectation	10.0 (5.0) 9.5 (2.7)
ESSA—Despondency	8.0 (4.0) 8.0 (2.3)
STAI—State	44.0 (25.0) 44.0 (15.7)
STAI—Trait	45.0 (22.0) 44.5 (12.7)
Self-esteem Scale	28. (6.0) 27.4 (5.3)
ASQ—Stress of Home Life	36.0 (16.0) 35.8 (10.8)
ASQ—Stress of School Performance	23.0 (9.0) 22.4 (6.2)
ASQ—Stress of School Attendance	8.0 (5.0) 8.3 (3.3)
ASQ—Stress of Romantic Relationships	15.0 (10.0) 15.4 (6.1)
ASQ—Stress of Peer Pressure	20.0 (13.0) 20.5 (7.7)
ASQ—Stress of Teacher Interaction	18.0 (10.0) 18.8 (7.1)
ASQ—Stress of Future Uncertainty	11.0 (4.0) 10.9 (3.1)
ASQ—Stress of School/Leisure Conflict	18.0 (8.0) 17.1 (4.7)
ASQ—Stress of Financial Pressure	12.0 (6.0) 11.9 (4.3)
ASQ—Stress of Emerging Adult Responsibility	8.0 (4.0) 8.1 (3.1)

**Table 3 ijerph-20-04692-t003:** Differences in ESSA scales by selected demographic characteristics and correlations (Spearman rho) with the other study measurements.

	Categories	Self-Expectation	Workload	Despondency	Pressure from Study	Worry about Grades
**Sex** Median (IQR) Mean (SD)	Males	8.0 (3.8) 8.6 (2.4)	9.0 (4.0) 9.4 (2.4)	7.0 (3.0) 7.4 (2.2)	11.0 (4.0) 11.1 (3.0)	8.0 (4.0) 7.9 (2.2)
	Females	10.0 (4.0) 10.0 (2.7)	11.0 (4.0) 10.2 (2.5)	8.0 (3.0) 8.4 (2.2)	13.0 (3.0) 12.5 (2.8)	9.0 (3.0) 8.7 (2.6)
	*p*-value	<0.0001	0.002	0.014	<0.0001	<0.0001
**Age (years)**	Spearman’s rho	0.180	0.198	0.186	0.220	0.098
	*p*-value	<0.001	<0.001	<0.001	<0.001	0.051
**Siblings** Median (IQR) Mean (SD)	Yes	10.0 (5.0) 9.5 (2.7)	10.0 (4.0) 9.9 (2.5)	8.0 (3.0) 7.9 (2.3)	12.0 (4.0) 11.9 (2.9)	8.0 (4.0) 8.2 (2.3)
	No	10.0 (4.0) 9.5 (2.5)	10.0 (3.0) 9.7 (2.5)	8.0 (3.0) 8.2 (2.3)	12.0 (4.0) 12.4 (3.0)	9.0 (4.0) 9.0 (2.9)
	*p*-value	0.296	0.296	0.017	0.891	0.259
**Parents’ Marital Status** Median (IQR) Mean (SD)	Married	10.0 (5.0) 9.5 (2.8)	10.0 (4.0) 9.9 (2.5)	8.0 (3.0) 7.8 (2.3)	12.0 (4.0) 11.9 (3.0)	8.0 (4.0) 8.2 (2.4)
	Divorced/separated	10.0 (4.5) 9.5 (2.4)	10.0 (4.0) 10.0 (2.5)	8.5 (3.0) 8.5 (2.0)	12.0 (4.0) 12.1 (3.0)	9.0 (4.0) 8.7 (2.7)
	Death of one or both parents	9.5 (3.5) 8.9 (2.1)	11.0 (2.0) 10.8 (1.5)	9.0 (2.0) 9.3 (2.3)	12.5 (4.8) 12.6 (2.6)	9.5 (3.8) 9.6 (1.9)
	*p*-value	0.772	0.576	0.144	0.792	0.004
**Parents’ Educational Status** Median (IQR) Mean (SD)	Until upper secondary school (Lyceum)	10.0 (4.0) 9.4 (2.7)	10.0 (4.0) 9.9 (2.3)	8.0 (4.0) 8.0 (2.2)	12.0 (4.0) 11.7 (2.8)	9.0 (4.0) 8.5 (2.5)
	Bachelor	10.0 (4.0) 9.3 (2.6)	10.0 (4.0) 9.9 (2.6)	8.0 (3.0) 7.9 (2.3)	12.0 (4.0) 12.0 (3.1)	8.0 (4.0) 8.3 (2.5)
	MSc/Ph.D.	11.0 (5.0) 10.0 (2.9)	10.0 (4.0) 9.8 (2.6)	8.0 (3.0) 8.2 (2.4)	13.0 (5.0) 12.6 (3.0)	9.0 (3.0) 8.4 (2.2)
	*p*-value	0.120	0.908	0.508	0.174	0.498
**Parents’ Work Status** Median (IQR) Mean (SD)	Unskilled	10.5 (5.0) 9.5 (2.4)	10.0 (4.3) 9.8 (2.4)	8.0 (4.3) 8.4 (2.1)	13.0 (4.0) 11.9 (2.9)	10.0 (4.5) 10.1 (3.0)
	Semi-skilled	10.0 (4.8) 9.8 (2.3)	11.0 (3.0) 10.3 (2.3)	8.5 (2.0) 8.2 (2.0)	13.0 (4.8) 12.6 (3.0)	9.0 (3.0) 8.8 (1.8)
	Skilled	10.0 (5.0) 9.4 (2.7)	10.0 (4.0) 9.9 (2.5)	8.0 (4.0) 8.0 (2.3)	12.0 (4.0) 11.9 (3.0)	8.0 (4.0) 8.2 (2.4)
	*p*-value	0.365	0.681	0.007	0.775	0.369
**Time Spent on Schoolwork Daily;** Median (IQR) Mean (SD)	Less than 2 h	10.0 (5.0) 9.7 (2.9)	9.0 (5.0) 9.2 (2.7)	8.0 (4.0) 8.2 (2.6)	11.0 (4.0) 11.2 (3.1)	8.0 (4.0) 8.4 (2.6)
	2–3 h	9.0 (4.0) 8.7 (2.7)	9.0 (4.3) 9.3 (2.5)	7.0 (3.0) 7.6 (2.3)	11.0 (4.0) 11.2 (2.9)	8.0 (3.0) 8.0 (2.4)
	More than 3 h	11.0 (4.0) 10.3 (2.3)	11.0 (2.0) 11.1 (1.9)	8.0 (3.0) 8.4 (1.9)	14.0 (3.0) 13.4 (2.4)	9.0 (3.0) 8.9 (2.5)
	*p*-value	<0.0001	<0.0001	0.003	<0.0001	0.002
**Private Tutoring During the Last Year;** Median (IQR) Mean (SD)	Yes	10.0 (5.0) 9.6 (2.6)	10.0 (3.0) 10.1 (2.5)	8.0 (4.0) 8.1 (2.2)	12.0 (4.0) 12.2 (2.8)	9.0 (3.0) 8.5 (2.5)
	No	7.0 (4.3) 8.3 (2.9)	8.0 (4.0) 8.3 (2.4)	7.0 (2.3) 7.2 (2.3)	10.0 (5.0) 10.3 (3.2)	7.0 (3.0) 7.4 (2.3)
	*p*-value	<0.0001	<0.0001	0.001	0.001	0.005
**Class During Weekends or Holidays;** Median (IQR) Mean (SD)	Yes	10.5 (4.0) 10.0 (2.6)	11.0 (3.0) 10.5 (2.4)	8.0 (3.8) 8.3 (2.3)	13.0 (4.0) 12.8 (2.9)	9.0 (4.0) 9.0 (2.5)
	No	9.0 (4.0) 8.9 (2.7)	9.0 (4.0) 9.3 (2.5)	8.0 (3.0) 7.7 (2.2)	11.0 (4.0) 11.2 (2.8)	7.0 (3.0) 7.8 (2.2)
	*p*-value	<0.0001	<0.0001	<0.0001	<0.0001	<0.0001
**Grades During Last Year;** Median (IQR) Mean (SD)	Low	8.0 (1.5) 8.0 (1.6)	10.0 (2.0) 9.9 (1.7)	10.0 (3.5) 10.0 (2.1)	11.0 (2.0) 11.7 (2.1)	10.0 (4.0) 9.9 (2.0)
	Medium	10.0 (4.0) 9.7 (2.6)	10.0 (4.0) 9.9 (2.7)	9.0 (4.0) 9.1 (2.3)	12.0 (4.0) 12.0 (3.0)	9.0 (4.0) 9.0 (2.7)
	High	9.0 (4.0) 9.4 (2.5)	10.0 (4.0) 9.8 (2.5)	8.0 (3.0) 7.5 (1.8)	12.0 (4.0) 12.0 (2.9)	8.0 (3.0) 8.1 (2.2)
	Very high	10.0 (6.0) 9.7 (3.4)	10.0 (4.0) 10.1 (2.6)	7.0 (3.0) 6.9 (2.3)	12.0 (4.0) 11.8 (3.4)	8.0 (4.0) 7.9 (2.6)
	*p*-value	0.917	0.888	<0.0001	0.160	<0.0001
**Health State During the Last 30 days;** Median (IQR) Mean (SD)	Bad	11.0 (2.5) 11.7 (1.5)	12.0 (4.5) 11.9 (2.6)	11.0 (5.5) 9.8 (3.4)	15.0 (4.5) 14.0 (2.8)	8.0 (3.0) 8.4 (1.8)
	Average	11.0 (2.8) 10.8 (2.4)	11.0 (3.0) 10.6 (2.5)	8.0 (4.0) 8.4 (2.3)	13.0 (3.0) 13.1 (2.8)	9.0 (3.8) 9.2 (2.9)
	Good	9.0 (4.0) 9.0 (2.6)	10.0 (4.0) 9.6 (2.5)	8.0 (3.0) 7.8 (2.2)	12.0 (4.0) 11.6 (2.9)	8.0 (4.0) 8.1 (2.2)
	*p*-value	<0.0001	<0.0001	0.002	<0.0001	0.020
**Physical Exercise During the Last 30 Days;** Median (IQR) Mean (SD)	Yes	9.0 (4.0) 9.3 (2.7)	10.0 (4.0) 9.7 (2.4)	8.0 (3.5) 8.0 (2.2)	12.0 (4.0) 11.9 (2.9)	8.0 (4.0) 8.2 (2.3)
	No	10.0 (4.0) 9.9 (2.7)	11.0 (3.3) 10.6 (2.7)	8.0 (4.0) 8.1 (2.4)	13.0 (4.3) 12.4 (3.2)	9.0 (4.0) 9.1 (2.9)
	*p*-value	0.115	0.001	0.019	0.046	0.596
**Use of Internet During the Last 30 Days;** Median (IQR) Mean (SD)	Yes	10.0 (9.3) 9.5 (2.7)	10.0 (9.5) 9.9 (2.5)	8.0 (7.7) 8.0 (2.3)	12.0 (11.0) 12.0 (3.0)	8.0 (7.5) 8.4 (2.5)
	No	12.0 (0.0) 12.0 (.)	11.0 (0.0) 11.0 (.)	10.0 (0.0) 10.0 (.)	15.0 (0.0) 15.0 (0.0)	10.0 (0.0) 10.0 (0.0)
	*p*-value	0.230	0.697	0.381	0.286	0.307
**Electronic Games During the Last 30 Days** Median (IQR) Mean (SD)	Yes	9.0 (5.0) 9.3 (2.8)	9.0 (4.0) 9.5 (2.7)	8.0 (4.0) 7.9 (2.5)	12.0 (4.0) 11.4 (3.1)	8.0 (4.0) 8.2 (2.5)
	No	10.0 (3.0) 9.6 (2.5)	11.0 (3.0) 10.4 (2.2)	8.0 (2.5) 8.2 (1.9)	13.0 (4.0) 12.7 (2.6)	9.0 (3.0) 8.6 (2.5)
	*p*-value	<0.0001	<0.0001	0.057	0.217	0.118
**ESSA—Total Score**	Spearman’s rho	0.793	0.676	0.750	0.874	0.744
	*p*-value	<0.001	<0.001	<0.001	<0.001	<0.001
**ESSA—Pressure from Study**	Spearman’s rho	0.627	0.596	0.564	1.000	0.573
	*p*-value	<0.001	<0.001	<0.001	-	<0.001
**ESSA—Workload**	Spearman’s rho	0.420	1.000	0.324	0.596	0.339
	*p*-value	<0.001	-	<0.001	<0.001	<0.001
**ESSA—Worry about Grades**	Spearman’s rho	0.487	0.339	0.548	0.573	1.000
	*p*-value	<0.001	<0.001	<0.001	<0.001	-
**ESSA—Self-expectation**	Spearman’s rho	1.000	0.420	0.526	0.627	0.487
	*p*-value	-	<0.001	<0.001	<0.001	<0.001
**ESSA—Despondency**	Spearman’s rho	0.526	0.324	1.000	0.564	0.548
	*p*-value	<0.001	<0.001	-	<0.001	<0.001
**STAI—State**	Spearman’s rho	0.627	0.337	0.478	0.524	0.417
	*p*-value	<0.001	<0.001	<0.001	<0.001	<0.001
**STAI—Trait**	Spearman’s rho	0.668	0.364	0.563	0.536	0.508
	*p*-value	<0.001	<0.001	<0.001	<0.001	<0.001
**Self-esteem Scale**	Spearman’s rho	−0.572	−0.270	−0.574	−0.431	−0.412
	*p*-value	<0.001	<0.001	<0.001	<0.001	<0.001
**ASQ—Stress of Home Life**	Spearman’s rho	0.503	0.336	0.579	0.617	0.473
	*p*-value	<0.001	<0.001	<0.001	<0.001	<0.001
**ASQ—Stress of School Performance**	Spearman’s rho	0.546	0.544	0.534	0.694	0.424
	*p*-value	<0.001	<0.001	<0.001	<0.001	<0.001
**ASQ—Stress of School Attendance**	Spearman’s rho	0.282	0.367	0.387	0.450	0.251
	*p*-value	<0.001	<0.001	<0.001	<0.001	<0.001
**ASQ—Stress of Romantic Relationships**	Spearman’s rho	0.232	0.147	0.392	0.357	0.271
	*p*-value	<0.001	0.003	<0.001	<0.001	<0.001
**ASQ—Stress of Peer Pressure**	Spearman’s rho	0.311	0.234	0.486	0.463	0.349
	*p*-value	<0.001	<0.001	<0.001	<0.001	<0.001
**ASQ—Stress of Teacher Interaction**	Spearman’s rho	0.308	0.284	0.484	0.447	0.357
	*p*-value	<0.001	<0.001	<0.001	<0.001	<0.001
**ASQ—Stress of Future Uncertainty**	Spearman’s rho	0.593	0.393	0.453	0.591	0.367
	*p*-value	<0.001	<0.001	<0.001	<0.001	<0.001
**ASQ—Stress of School/Leisure Conflict**	Spearman’s rho	0.458	0.430	0.416	0.584	0.307
	*p*-value	<0.001	<0.001	<0.001	<0.001	<0.001
**ASQ—Stress of Financial Pressure**	Spearman’s rho	0.377	0.270	0.429	0.481	0.279
	*p*-value	<0.001	<0.001	<0.001	<0.001	<0.001
**ASQ—Stress of Emerging Adult Responsibility**	Spearman’s rho	0.260	0.261	0.290	0.381	0.226
	*p*-value	<0.001	<0.001	<0.001	<0.001	<0.001

## Data Availability

Data are available upon request from the corresponding author.

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
