# Peer review of "Educational Stress among Greek Adolescents: Associations between Individual, Study and School-Related Factors"

_ijerph, 2023, doi:10.3390/ijerph20064692_

Round 1

Reviewer 1 Report

The manuscript addresses a very interesting international topic that can help to reduce students stress level and improve academic performance, under the title " Educational Stress among Greek adolescents: Associations between individual, study and school- related factors.".

1.      The topic of the manuscript, as well as the updated literature review provided by the authors, seems to be appropriate but is neither comprehensive nor current.  This section could be significantly improved by adding high impact scientific articles. In addition, I suggest some others points that could be improved.

2.      METHOD. The research design is correct, and the data used seem useful for the objectives set by the author of the text. Regarding Research Context and Participants. Some information about data collection process is necessary and change table of the sample into the Method Section.

Furthermore, it is not reported whether the sample analysed is representative, while the authors generalise the conclusions to the whole population analysed.

3.      RESULTS. I consider that this section needs a major revision.

The statistical tools used are simple and the analysis of the results is scarce and shallow. There are also many tables and little explanation of the results.

4.      CONCLUSION

This section needs a major revision. Authors don´t shows the main conclusions of the present study.  It must be mentioned the limitations and future lines of the investigations.

5.      Format error

a.       Remove space. Lines 58,64,68, 199… Please revise all the document

b.      Citation format line 228

I hope my suggestions will be useful.

Author Response

Dear Reviewer,

Thank you for comments and suggestions. We are taking into consideration those points. 

More details could be found in the attachment.

Reviewer 2 Report

Dear authors,

Some modifications are recommended to increase the final quality:

1.- The article must be completed with a theoretical framework and previous research.

2.- Well described and structured quantitative methodology, although it is recommended to make the questions and objectives more explicit. Also, I have some questions around your methodology: 1.- Why did you choose this methodological approach? 2.- How was the data analysed?

3.- The results chapter is relevant, but it would be recommended that the topics / categories in which the analysis and interpretation are structured be justified as they emerge, either in this results section, or preferably in the method section as indicated in the previous comment.

4.- The authors should point out the main results of the study, the limitations, and some ideas for future research.

5.- Regarding the current conclusions section, it is recommended to rewrite the contents.

6.- Finally, it is necessary to review the bibliography and extend the references.

Kind regards

Author Response

Dear reviewer,

Thank you for your comments and accurate points. Now we recommend some limitations and we suggest points for future studies.

Reviewer 3 Report

The survey and analysis are well conducted, and the conclusions are supported by the empirical results. Moreover, the subject (educational stress) is important and widely discussed. 

I have some minor issues and concerns, primarily concerning a lack of self-critical discussion:

- The number of respondents is quite small (N = 399, most respondents from Athens). Is that a concern?

- What is the novelty of the study compared with other studies of educational stress? Most (if not all) results are in line with the existing literature.

- Finally, I wonder, whether the situation in Greece concerning educational stress differs from the situation in other, comparable countries? Is the "study culture" in Greece different from the study culture in e.g. Nordic countries, the UK, etc.? And could we learn something from such potential differences?

Author Response

Dear reviewer,

Thank you for your accurate suggestions. there are some limitations about the sample. It is a major concern but we did not have the opportunity to include sample from other areas in Greece.

We recommend the study culture in Greece after your point.  More details could be found in the attachment.

Reviewer 4 Report

Thank you for the opportunity to review the manuscript entitled "Educational stress among Greek adolescents: Associations between individual, study, and school-related factors".  The authors use data obtained by self-report to examine relationships between individual and contextual characteristics (i.e., age, gender, academic behaviors, parental occupation) and mental health outcomes (such as stress and anxiety).  Several relationships are identified, and implications for intervention are discussed.  Please see below for several points of feedback.

Stress varies tremendously across different contexts.  I presume that the sample is not nationally representative of adolescents in Greece, although the authors do not state this.  It is imperative to discuss the diversity within the sample (with regard to socioeconomic status, membership to marginalized communities, etc.).  Stressors vary according to these dimensions, as do perceptions of academic stressors, and there is no information to contextualize the findings.  The authors perhaps touch on these points as they classify parental work status, but more information is needed.

The authors measure both general stress (via the Adolescent Stress Questionnaire) and educational stress specifically.  More information is needed about these two constructs; how they potentially overlap (and how this might impact observed results) and how they differentially predict or are predicted by other variables.

Perhaps most importantly, as the authors acknowledge this is a cross-sectional study and thus the ability for the authors to draw causal inferences is very limited.  For example, it is impossible to ascertain whether extra study hours is a source of stress, and / or whether students who are predisposed to stress and anxiety feel extra pressure to devote time to studying.  It is likely that both pathways are valid.  However, given the inability to make causal inferences, and the lack of novelty (in constructs, measurement approaches, or other aspect), I question the importance of these findings.

There is an error message on line 121.

There is a type on line 154.

Author Response

Dear reviewer, 

Thank you for your suggestions and accurate points. 

The information was collected relying on self-reports measurements and some recall bias cannot be avoided. The relationships between factors cannnot be interpreted causally because of the cross-sectional nature of this study. The sample was chosen conveniently and the findings cannot be generalized to the population.

More details could be found in the attachment.

Round 2

Reviewer 1 Report

The authors have provided few changes to the document. However, the paper can be published

Author Response

Thank you for your accurate comments

Reviewer 4 Report

This manuscript is improved; however, I still think it requires further justification for the utility or novelty of the findings.  For example, the correlations between study hours, private tutoring, and stress - there is likely a degree of selection occurring, where students who are predisposed to stress seek more study hours and tutoring.  Simultaneously, parents who encourage more study hours and tutoring might place more pressure on their children, thereby generating more stress.  Since the directionality can't be determined (as the authors acknowledge), there needs to be more discussion about how to apply these findings (which aren't particularly surprising).  The new use of the measures in the context of Greece is important, perhaps this could be further emphasized.  The gender differences are also interesting and could be further discussed (i.e., implications, comparisons with other countries, etc.).  Please also see below for specific points of correction.

There is an error on line 145.

There is a type on line 178.

Author Response

Thanks for accurate comments. We mentioned some points for gender differences. Also, thanks for specific points of correction. We applied these findings to intervention programs in future research.